# The development and implementation of a community engagement strategy to improve maternal health in southern Mozambique

Felizarda Amosse[1], Mai-Lei Woo Kinshella[2], Helena Boene[1], Sumedha Sharma[2], Zefanias Nhamirre[1], Corssino Tchavana[1], Laura A. Magee[2,3], Peter von Dadelszen[2,3], Esperança Sevene[1,4], Marianne Vidler[2], Khatia Munguambe[1,4]*, the CLIP Mozambique Working Group[¶]

**1** Centro de Investigação em Saúde da Manhiça, Maputo, Mozambique, **2** Department of Obstetrics and Gynaecology, University of British Columbia, Vancouver, British Columbia, Canada, **3** Department of Women & Children's Health, King's Collage London, London, United Kingdom, **4** Faculdade de Medicina, Universidade Eduardo Mondlane, Maputo, Mozambique

☯ These authors contributed equally to this work.
¶ Membership of the CLIP Working Group is provided in the Acknowledgments
* khatia.munguambe@manhica.net

**Data Availability Statement:** The data are available upon request to the CISM's institutional ethics committee (sozinho.acacio@manhica.net) and the Mozambican Ministry of Health National Bioethics

## Abstract

Delays to seek medical help can contribute to maternal deaths particularly in community settings at home or on the road to a health facility. Community engagement (CE) can improve care-seeking behaviours and complements community-based interventions strengthening maternal health. The purpose of this paper is to describe the process undertaken to develop and implement a large-scale community engagement strategy in rural southern Mozambique. The CE strategy was developed within the context of the "Community-Level Interventions for Pre-eclampsia" (NCT01911494) conducted between 2015–2017 in southern Mozambique. Key CE messages included pregnancy complications and their warning signs, including pre-eclampsia and eclampsia, as well as emergency readiness, birth preparedness, decision-making mechanisms, transport options and information about the trial. CE meeting logs were used to record quantitative and qualitative information on demographic data and feedback. Quantitative data was analyzed using RStudio (RStudio Inc, Boston, United States) and community feedback was qualitatively analyzed on NVivo12 (QSR International, Melbourne, Australia). CE activities reached 19,169 participants during 4,239 meetings. CE activities were reported to be well received by community members though there was a relatively lower participation of men (3565 /18.6%). The use of recognized local leaders and personnel, such as community leaders, nurses and community health workers, allowed for greater acceptance of CE activities and maximized coverage of health messages in the community setting. Our CE strategy was effective in integrating maternal health promoting activities in routine care of community health workers and nurses in the area. Understanding district differences, engaging husbands, partners, mothers-in-law and community-level decision-makers to build local support for maternal health and flexibility to tailor messages to local needs were important in developing sustainable forms of

Committee (jflschwalbach@gmail.com) for researchers who meet the criteria to access confidential data.

**Funding:** This work is part of the University of British Columbia PRE-EMPT (Pre-eclampsia/ Eclampsia, Monitoring, Prevention and Treatment) initiative supported by the Bill & Melinda Gates Foundation (Grant number: OPP1017337, PvD). Following input into trial design, the Gates Foundation had no role in data collection, analysis, or interpretation, or writing of the report.

**Competing interests:** The authors have declared that no competing interests exist.

CE. Better strategies are needed to effectively engage men in maternal health promotion who were less available due to working outside of the home or neighbourhoods

## Introduction

Similar to many countries in sub-Saharan Africa, most maternal and neonatal deaths in Mozambique occur during childbirth and the first few days of life. Lack of access to appropriate medical care contributes significantly to these deaths [1]. A study on the prevalence of the use of institutional delivery services in Mozambique showed that 29% of deliveries occurred outside of health facilities and the proportion of home births was four times higher in rural areas (40%) than in urban areas (10.8%) [2], suggesting sub-optimal health care seeking behaviours [3] and poor distribution of services [4]. The delay to decide to seek medical help is a major contributor to many deaths that occur at home or on the road to a health facility [5]. Appropriate care-seeking behaviour involves first recognizing the danger signs for women and newborns. Knowledge of danger signs in pregnancy, pre-eclampsia in particular, are limited and represent a challenge in the reduction of maternal and infant mortality. A study in southern Mozambique showed that rural communities had little knowledge about pregnancy complications, especially about the origin of pre-eclampsia [6]. Faced with challenges related to access and knowledge, community members may adopt use of traditional beliefs and practices, some of which may carry higher risks to maternal and child health [6,7].

Community engagement (CE) can improve care-seeking behaviours and is an essential part of implementing sustainable health promotion programs [8]. CE in health interventions strengthens communities through increasing their self-confidence, self-awareness and self-reflection, which supports finding local solutions to health problems [9]. Consequently, the World Health Organization (WHO) recognizes the importance of engaging communities in promoting health and describes CE as vital to linking health problems to appropriate health promotion actions [10]. The WHO highlights that CE can improve maternal and newborn health behaviours, increase use of skilled care and increase household and community support for maternal and newborn health [11]. However, there is a gap in identifying sustainable forms of CE [9]. In order to contribute to a better understanding of locally-tailored forms of engaging communities, including the sustainability of such approaches, this paper describes the process of development and implementation of a large-scale CE strategy in rural Mozambique.

## Methods

### Study design and areas

The CE strategy was developed and implemented as an integral part of the "Community-Level interventions for Pre-eclampsia (CLIP) in southern Mozambique: a cluster randomized controlled trial" (NCT01911494) conducted between 2015–2017. CLIP was conducted as a collaboration between the University of British Columbia (UBC) in Canada and Centro de Investigação em Saúde da Manhiça (CISM) and Universidade Eduardo Mondlane (UEM) in Mozambique. The objective of the CLIP Trial was to reduce maternal, perinatal and neonatal mortality and morbidity through a community-based intervention building capacity of community health workers (CHWs), referred to as *Agentes Polivalentes Elementares* (APE) in Mozambique. These CHWs worked in close collaboration with primary health care nurses [12,13].

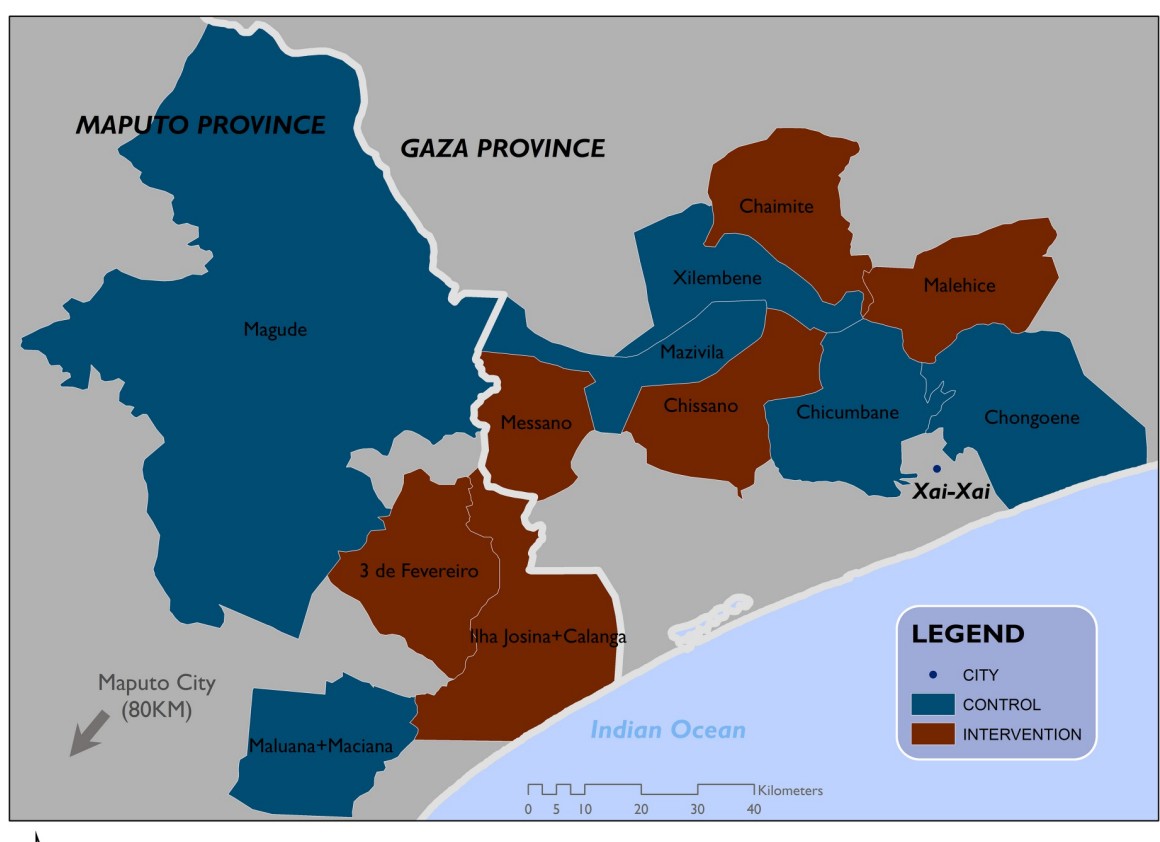

**Intervention and Control clusters for the CLIP Mozambique trial**

**Fig 1. Study area map.**

The CE strategy was implemented in six regions: two from Manhiça district in Maputo province and four within Bilene Macia and Chibuto districts in Gaza province (**Fig 1**). The six regions were selected because they constituted the intervention areas of the trial. All three districts are rural, they ranged in size from 1,289.8 to 1,829.1 km² with a population of 7,197 to 16,129 (**Table 1**). Residents belong to the *Xichangana* ethnic group. Agriculture represents the economic basis for families' subsistence and the main cash crops are sugarcane, cotton and tobacco.

## Community engagement strategy development

CE is a process that encourages community participation in activities to improve population health and/or reduce health inequalities [14]. The WHO defines CE as, "a process of

**Table 1. District demographics.**

| Province | District | Administrative Post | Area (km²) | Population | Number of households |
|---|---|---|---|---|---|
| Maputo | Manhiça | Calanga and Ilha Josina | 1079.4 | 18,870 | 1,317 |
| | | Três de Fevereiro | 749.7 | 40,208 | 5,880 |
| Gaza | Bilene Macia | Messano | 573.4 | 23.921 | 4,716 |
| | | Chissano | 716.4 | 61.165 | 11,413 |
| | Chibuto | Malehice | 816.0 | 65,451 | 4,107 |
| | | Chaimite | 838.9 | 27,863 | 4,249 |

**Table 2. Overview of the community engagement development and implementation plan.**

| Identified barriers and needs | Community engagement strategy | Implemented community engagement activities |
|---|---|---|
| 1) Poor awareness of pre-eclampsia, eclampsia and pregnancy complication warning signs | Develop and deliver educational messages and visual materials about pregnancy warning signs and care seeking behaviours | Three forms of education materials created, tested and used: Q&A booklets, warning signs flipchart, key message guide for facilitators. 48 CHW engage in health promotion using purpose made materials in house visits targeting women, husbands/ partners, and other family decision-makers. |
| 2) Late disclosure of pregnancy/delayed first antenatal visit | Develop and deliver educational messages about the benefits of early antenatal care and discuss case scenarios at meetings | 16 nurses engage in PHC health talks, using purpose made materials targeting women in reproductive age and Pregnant women, and their escorts |
| 3) Women´s low decision making power | Involve male partners, in-laws and community-level decision-makers as CE target groups  Discuss decision-making case scenarios, emergency readiness and birth preparedness | 5 activists and 3 mobilizers, supported by community leaders, facilitate community meetings covering all CE target groups |
| 4) Community transport limitations | Development of an emergency transportation strategy linked to community savings schemes | Co-creation an implementation of a community transport scheme at neighbourhood level based on demand |

developing relationships that enable stakeholders to work together to address health-related issues and promote well-being to achieve positive health impact and outcomes" [14]. CE can support building trust, enlisting new resources and allies, and improving communication to support environmental and behavioral changes that will improve the health of the community and its members [15].

Our CE strategy was developed based on the results obtained in a feasibility study to understand women's health care seeking practices during pregnancy and identify underlying social, cultural and structural barriers to accessing timely appropriate care in CLIP Trial communities [16]. Through in-depth interview and focus group discussions with women of reproductive age (WRA), pregnant women, household decision makers, formal and informal health care providers, local health authorities, community leaders and other influencers, the feasibility study identified a number of maternal health barriers that formed topics to be addressed by our CE strategy. These included: 1) Poor awareness of pre-eclampsia, eclampsia and pregnancy complication warning signs; 2) Late disclosure of pregnancy, leading to delayed first antenatal care visit; 3) Women´s low decision making power; and 4) Community transport limitations (**Table 2**). Therefore, the CE strategy implementation first aimed to encourage community dialogue on obstetric complications and healthcare-seeking practices. Secondly, as the feasibility study also identified local contextual factors that influence capacity for care-seeking, these were considered by different accompanying components of the PRE-EMPT project. For instance, because lack of transport was identified as a major contributor to poor referrals and represented a barrier to adequate maternal healthcare access [16], a community transport strategy was also developed to leverage existing community resources within a group savings scheme. The community transport strategy is reported further in Amosse et al 2021 [17].

Throughout the development process, we consulted community members, APEs, health care providers at health facilities and the Mozambican Ministry of health regarding our proposed CE strategy to ensure it was locally feasible and appropriate.

## Community engagement personnel and activities

CE activities took place in three spaces: primary health centers (PHCs), in the homes of pregnant and puerperal women and in community meetings. CE activities were conducted by three groups of facilitators: 16 nurses conducted activities at PHCs, 48 CHWs during home visits, and five activists and three mobilizers at community meetings. Maternal and child health nurses provide daily health talks as outlined by the Ministry of Health at PHCs and messages regarding pre-eclampsia, eclampsia and other topics were added to these daily talks (**Box 1**). Daily health talks largely targeted pregnant women and WRA, but also included

husbands and other relatives. CHWs are community members selected and trained by the Ministry of Health to provide basic medical care at the community level. For CLIP Trial, they were also trained to recruit pregnant women, conduct follow-up visits, provide messages regarding pre-eclampsia and eclampsia, assess their risk for pre-eclampsia and eclampsia and other complications and refer them to the nearest health facility when necessary (**Box 1**).

---

### Box 1. Topics covered at community engagement activities

1. Signs and symptoms of pregnancy complications, in particular preeclampsia and eclampsia

2. Permission for care seeking in pregnancy

3. Identification of skilled birth attendants

4. Identification of a facility for delivery

5. Transport and treatment funds: encourage the identification of existing community resources and the development of community funds for emergency care.

6. Feedback on adverse outcomes and 'great saves'

7. Study protocol overview*

8. Details on specific CLIP study interventions (pre-eclampsia diagnosis, treatment and referral)

9. Solutions to the previously identified barriers to maternal and neonatal health

* *Included discussion of the PRE-EMPT study objectives, target groups to be recruited and study procedures.*

---

Activists are community members that engaged with the CLIP team and were trained to facilitate community meetings in collaboration with community leaders, who acted as gate-keepers by scheduling and inviting people to participate in community meetings. Because they were based in the community and had a flexible availability, activists frequently conducted the CE meetings. Meetings were also sometimes conducted by mobilizers. Mobilizers are staff based at the central CISM office who made field visits three times a week to conduct community meetings and monitoring activities conducted by nurses, CHWs and activists.

Activities led by nurses and CHWs focused more on health oriented topics, while CISM field staff focused more on research and socio-cultural elements. Overall, facilitators were well trained, equipped with previous research experience, familiar with the community context, and fluent in the local language. Relationships with the communities were established prior to data collection by approaching the administrative post chiefs, traditional leaders, and the neighbourhood secretary for prior permission.

The entire team was coordinated by a Community Liaison Officer (CLO), who oversaw the activities, monitored progress, trouble-shot on operational problems, managed the data, and was the linkage with the investigators' team and the research centre.

### Materials

A number of materials were developed to support CE facilitators (nurses, CHW, activists and mobilizers). An informational booklet about pre-eclampsia was developed from questions

raised by community members during the feasibility study. Two thousand copies of this booklet were distributed to CE facilitators, community leaders, religious leaders, pregnant woman, WRA, partners and matrons. Pictogram flip chart, posters, and key message scripts which distilled clinical explanations into illustrations and lay language were also developed.

### Community engagement messages

The CE activities focused on the key messages regarding pre-eclampsia and eclampsia, pregnancy complications and care-seeking in pregnancy (**Box 1**).

### Data collection and management

CE logs were completed by facilitators after all CE sessions at the health facility, individual homes and community meetings. Information recorded included details on the region, target group, number of participants and messages discussed. CE logs also included community feedback notes where facilitators recorded brief reflections and comments from participants in free text format, which were used for the qualitative analyses. Mobilizers collected CE logs from the community and health facilities and centralized the logs at the CISM office for data entry. CE logs were monitored by mobilizers once a week and quality checked by community liaison officer once a month. While collecting the logs there were opportunities to verify the completeness of the data and clarify or resolve missing data or discrepancies. Missing or questionable data was queried by the study team for follow up. Logs were entered into a REDCap database (Vanderbilt University, Nashville, United States). Data entry from paper records to the electronic database was quality checked for accuracy.

### Data analysis

Descriptive statistics were conducted with the quantitative data using RStudio (RStudio Inc, Boston, United States) to compile frequency of demographic characteristics, number of meetings and the topics discussed. Data was then presented in the format of tables and bar charts, and frequencies were compared by region and by target group. These outputs were used to monitor CE activities and adjust when necessary. The community feedback notes from CE logs were imported to NVivo12 (QSR International, Melbourne, Australia) for content analysis. Conventional content analysis is an inductive method of qualitative analysis to interpret meaning from the content of data as expressed by study participants without imposing preconceived categories [18]. Conventional content analysis involves first familiarizing with the data and reading data word by word. Exact words from the data that capture key issues raised by participants are highlighted as codes. Codes are subsequently sorted into categories, which are used to organize codes into meaningful clusters of topics.

### Ethical considerations

Approval for this study was obtained from the Institutional Bioethics Research Boards of *Centro de Investigação em Saúde da Manhiça* (CISM, CIBS-CISM/038/14), the Mozambique National Bioethics for Health Committee (219/CNBS/14) and the University of British Columbia (UBC, H12-03497). Written informed consent was obtained from all households participating in the CLIP Trial, of which community engagement was a nested component. By giving consent to participate in the CLIP trail, participants also agreed for community engagement activities, which were only conducted in CLIP intervention clusters. Following the CLIP protocol [12], adolescents in the Trial gave assent combined with a parental or guardian consent. All CE records were anonymized.

# Results

## Participants' characteristics

During the two-year study, CE activities reached 19,169 participants between the ages of 12 and 96 years old (**Table 3**). Fifty-one percent of the participants were married or cohabiting with their partner. The majority of participants were women (81.4%), and most participants worked in subsistence farming (55.5%), while 18.7% were unemployed.

## Community engagement coverage

During CE activities, 4,239 meetings were conducted in Bilene Macia, Chibuto and Manhiça districts (**Table 4**). The Bilene Macia district had the highest number of meetings and participants (41.4% and 38.9% respectively) while Manhiça district had the fewest number of meetings and participants (27.3% and 22.3% respectively). There was a median [interquartile range] of 810.5 [464.8–909.5] meetings per region.

The primary target group comprised of pregnant women and WRA, who represented the proportion meetings. Pregnant women and WRA were engaged in more than half (69%) of all meetings held throughout the entire CE process (**Table 5**). Next were meetings with combined target groups, which represented 13% of meetings. Though the numbers of meetings with community leaders represented a small proportion of overall CE meetings (0.1%), they were engaged in an impactful role in community meetings as gatekeepers and opinion leaders.

**Table 3. Participant information by district.**

| | Manhiça (n = 4267) | | Bilene Macia (n = 7473) | | Chibuto (n = 7429) | | Total (n = 19169) | |
|---|---|---|---|---|---|---|---|---|
| | # | % | # | % | # | % | # | % |
| **Gender** | | | | | | | | |
| Male | 684 | 16.0 | 1491 | 20.0 | 1390 | 18.7 | 3565 | 18.6 |
| Female | 3583 | 84.0 | 5982 | 80.1 | 6039 | 81.3 | 15604 | 81.4 |
| **Age** | | | | | | | | |
| Adolescents (12–19 years) | 1003 | 23.5 | 1429 | 19.1 | 1267 | 17.1 | 3699 | 19.3 |
| Adults (20–24 years) | 853 | 20.0 | 1342 | 18.0 | 1125 | 15.1 | 3320 | 17.3 |
| Adults (25–34 years) | 1265 | 29.6 | 2194 | 29.4 | 1926 | 25.9 | 5385 | 28.1 |
| Adults (35–49 years) | 732 | 17.2 | 1533 | 20.5 | 1947 | 26.2 | 4212 | 22.0 |
| Adults (50 years and higher) | 414 | 9.7 | 975 | 13.1 | 1164 | 15.7 | 2553 | 13.3 |
| **Marital status** | | | | | | | | |
| Married or marital union | 2750 | 64.5 | 4266 | 57.1 | 2834 | 38.1 | 9850 | 51.4 |
| Divorced, separated or widowed | 323 | 7.6 | 438 | 5.9 | 433 | 5.8 | 1194 | 6.2 |
| Single | 1186 | 27.8 | 2752 | 36.8 | 4143 | 55.8 | 8081 | 42.2 |
| Other | 8 | 0.2 | 17 | 0.2 | 19 | 0.3 | 44 | 0.2 |
| **Occupation** | | | | | | | | |
| Farming (agriculture and/or livestock) | 1563 | 36.6 | 4458 | 59.7 | 4242 | 57.1 | 10263 | 53.5 |
| Service providers (transport, health, teacher) | 218 | 5.1 | 339 | 4.5 | 317 | 4.3 | 874 | 4.6 |
| Housekeeper | 8 | 0.2 | 325 | 4.4 | 795 | 10.7 | 1128 | 5.9 |
| Student | 144 | 3.4 | 600 | 8.0 | 575 | 7.7 | 1319 | 6.9 |
| Unemployed | 1020 | 23.9 | 1396 | 18.7 | 1169 | 15.7 | 3585 | 18.7 |
| Community health volunteers | 1 | 0.0 | 7 | 0.1 | 5 | 0.1 | 13 | 0.1 |
| Retired | 1 | 0.0 | 0 | 0.0 | 1 | 0.0 | 2 | 0.0 |
| Traditional healers | 6 | 0.1 | 11 | 0.2 | 13 | 0.2 | 30 | 0.2 |
| Other | 1306 | 30.6 | 337 | 4.5 | 312 | 4.2 | 1955 | 10.2 |

**Table 4. Community engagement meetings covered by administrative post.**

| District | Administrative Post | Number of meetings | Participants |
|---|---|---|---|
| Manhiça | Três de Fevereiro | 806 | 2644 |
| | Calanga and Ilha Josina | 351 | 1623 |
| | **District total:** | **1157 (27.3%)** | **4267 (22.3%)** |
| Bilene Macia | Chissano | 941 | 4267 |
| | Messano | 815 | 3206 |
| | **District total:** | **1756 (41.4%)** | **7473 (38.9%)** |
| Chibuto | Malehice | 1020 | 4335 |
| | Chaimite | 306 | 3094 |
| | **District total:** | **1326 (31.3%)** | **7429 (38.8%)** |
| **Overall total:** | | **4239 (100%)** | **19169 (100%)** |

Bilene Macia, was the district that reached the highest number of meetings while Manhiça registered the lowest number of meetings.

Fig 2 illustrates the frequency of messages disseminated. The most discussed topic was "danger signs and symptoms of pregnancy complications", particularly those related to pre-eclampsia and eclampsia. The second most frequent topic discussed was "identification of skilled birth attendants".

Message frequency varied by region (Fig 3). For example, signs and symptoms of pregnancy complications was more widespread in Manhiça district, while permissions for care seeking was more widely addressed in Bilene-Macia and Chibuto districts. As the study progressed and facilitators and participants became more familiar with the topics, facilitators adapted the focus of discussions to existing gaps in the local communities.

## Stakeholders' reflections on engagement activities in communities

**Reflections on community engagement format.** Community engagement meetings were designed to be an "exchange of ideas" and participants were encouraged to ask questions and

**Table 5. Number of meetings stratified by target groups in each district.**

| Target groups reached | | Number of meetings | | | Overall number of meetings | |
|---|---|---|---|---|---|---|
| | | Manhiça | Bilene Macia | Chibuto | Freq. | % |
| Primary target group | Pregnant women | 1006 | 1134 | 799 | 2939 | 69.3 |
| | WRA | 10 | 83 | 121 | 214 | 5.0 |
| Household-level decision makers | Mothers and mothers-in-law | 15 | 23 | 17 | 55 | 1.3 |
| | Husbands and partners | 25 | 58 | 22 | 105 | 2.5 |
| Community-level decision-makers | Community leaders | 2 | 3 | 1 | 6 | 0.1 |
| | Small entrepreneurs and neighborhoods secretaries | 2 | 5 | 2 | 9 | 0.2 |
| Health professionals | CHWs | 12 | 9 | 3 | 24 | 0.6 |
| | Nurses | 5 | 4 | 3 | 12 | 0.3 |
| Others | Theatre and dance cultural groups | - | 1 | 1 | 2 | 0.1 |
| | Matrons and elders | - | 3 | 5 | 8 | 0.2 |
| Meetings with combined target groups | WRA/pregnant women with husbands and partners | 2 | 28 | 6 | 36 | 0.9 |
| | WRA/pregnant women | 13 | 129 | 77 | 219 | 5.1 |
| | WRA/ pregnant women with mothers and mother in laws | - | 25 | 17 | 42 | 1.0 |
| | Multiple target groups concurrently | 65 | 251 | 252 | 568 | 13.4 |
| **Total** | | **1157** | **1756** | **1326** | **4239** | 100 |

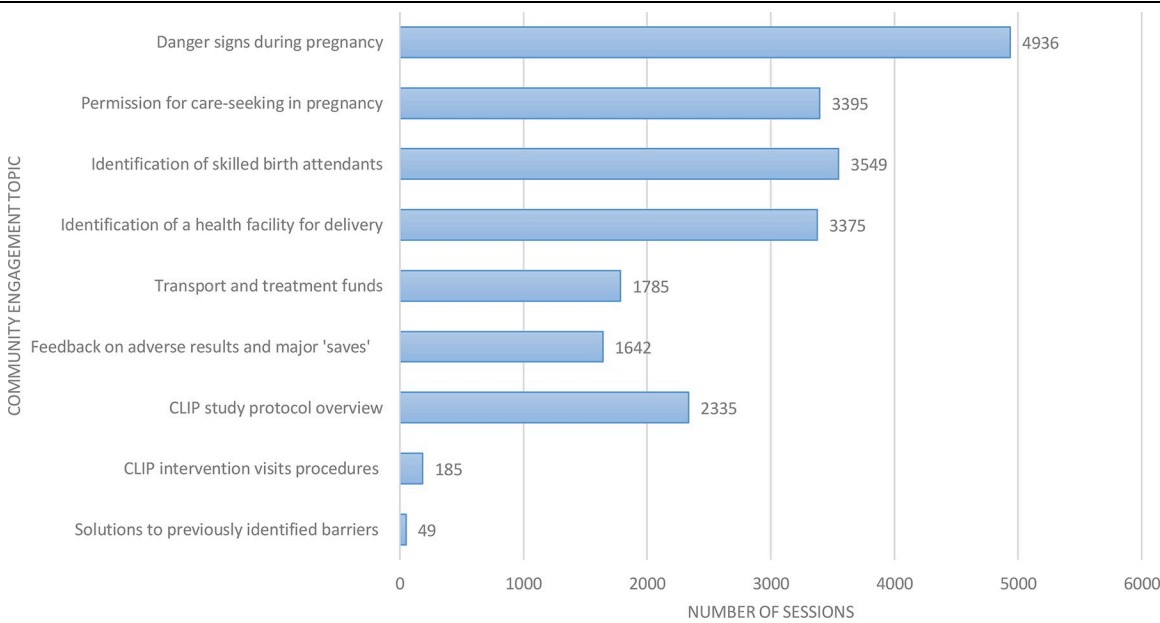

**Fig 2. Overall topics discussed during community engagement activities.**

share stories from their own communities. The CE activities included dedicated time for open questions and an opportunity to assess comprehension. The quote below from the community feedback illustrates the focus on dialogue and experience sharing to raise awareness of pre-eclampsia and eclampsia in the communities.

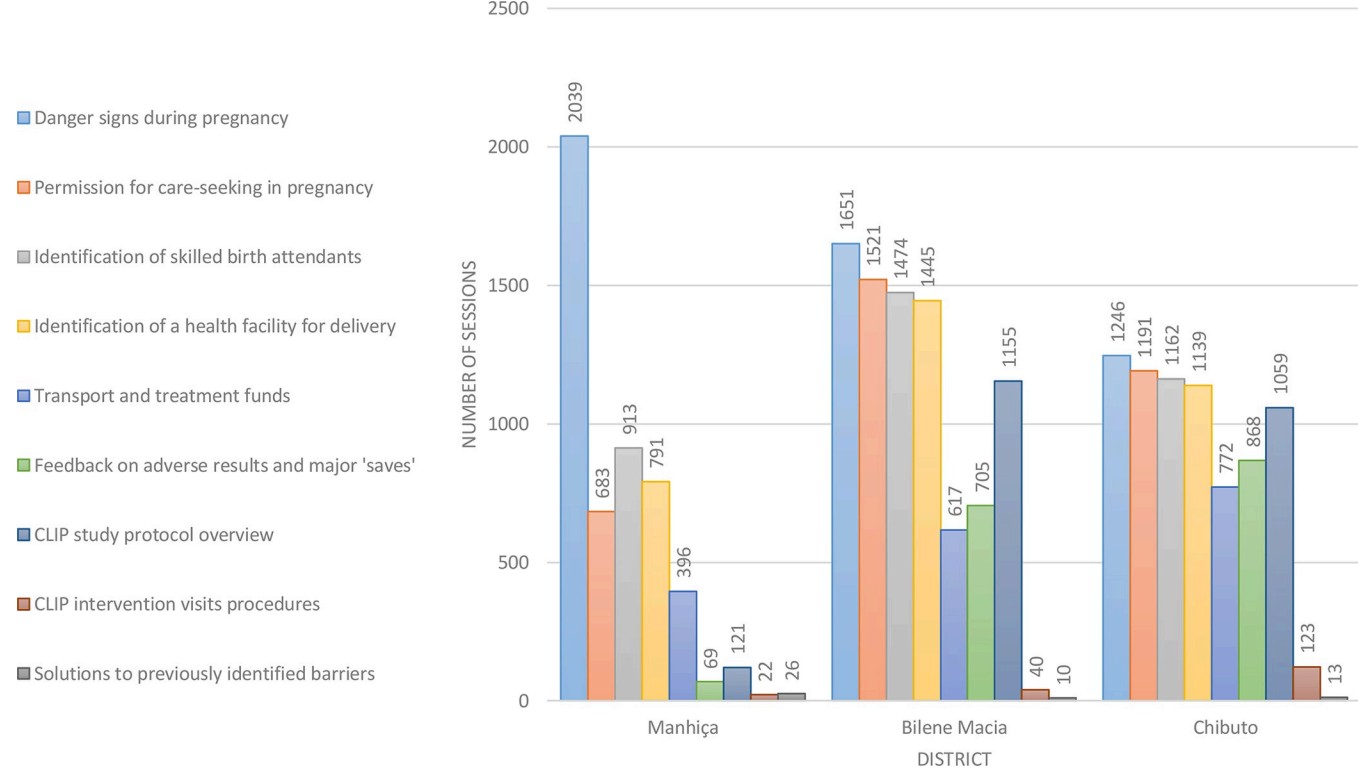

**Fig 3. Community engagement topics discussed by administrative post.**

"There was a big discussion about the disease (pre-eclampsia) taking into consideration that the participants in the debate were open and providing examples of cases experienced in their communities." Community leaders from Três de Fevereiro, Manhiça district

**Community reception of activities and messages.** A number of CE facilitators stated that participants seemed comfortable and engaged in discussions.

"I am very happy with this care (during the CLIP Trial) and lecture. I liked the information about the pre-eclampsia /eclampsia disease and danger signs in pregnant women." Pregnant woman from Malehice, Chibuto district

"The women listened attentively, they were curious and asked questions. . . They promised to go to the Health Facility whenever necessary or if they felt discomfort." Pregnant women from três de Fevereiro, Manhiça district

The use of recognized local leaders and personnel, such as community leaders, nurses and community health workers, allowed for greater acceptance of CE messaging. However, women were sometimes shy and reluctant to discuss openly, especially if a community leader was present. A few remarked that women were too tired to participate actively in CE activities.

"The pregnant woman's mother-in-law said she has no time to listen to what the CHW had to say because she was tired and hungry." A mother-in-law during a discussion with women with reproductive age in Chissano, Bilene Macia district

"In this neighborhood, the women were shy and didn't make many comments." Pregnant women from Ilha Josina, Manhiça district

While most feedback described positive community reception of activities and messages, CE facilitators reported some pushback from men who questioned the focus on maternal health and obstetric emergencies. They wondered why the CLIP Trial and CE messaging did not help them and were sometimes concerned that the messaging would increase fear among their wives and partners. While some men were critical of the project, many men were also positive about the project.

"We would like to understand why only the pregnant women are treated. Men die and the women in the community live." Husbands and partners from Messano, Bilene Macia district

"The partners were worried because they heard that this disease has no cure and becomes serious when not controlled. This will make women afraid to give birth." Husbands and partners from Chissano, Bilene Macia district

## Discussion

Community engagement activities during the CLIP Trial reached about 19,000 participants with over 4,000 meetings during the course of two years in rural communities in Southern Mozambique. Understanding local issues and barriers during the feasibility study allowed our team to develop a CE strategy that had a wide reach and was well received by community members. The focus on dialogue to deliver CE messaging helped to promote participant discussion on pre-eclampsia and eclampsia.

Lessons learned in developing sustainable forms of CE include understanding differences across districts, the importance of a continued dialogue and opportunities for feedback from communities, and the importance of flexibility to tailor messages to local needs and as participants became more familiar with the topics over duration of the study. Responsiveness to adapt CE strategies according to community needs and their responses to various study components has been identified as a key element of effective community engagement [19]. In our study, this was supported by encouraging CE facilitators' reflection on the process and key issues that emerged in each CE session, which provided a space to record feedback from communities, helped to strengthen facilitators' engagement with activities and supported monitoring and evaluation of the program. Facilitator reflections has been used in quality improvement initiatives to promote buy-in and motivation as well as provide the time and structure to reflect on potential adaptations to improve engagement [20,21].

Formative research has also been identified as a key element of effective community engagement to gain insight into the local socio-economic context and health needs [19]. Formative research can also support understanding differences between regions. Looking at the CE activities, we can see different dynamics of activities by district. Bilene Macia is a larger district in terms of population size (see Table 1) and a larger team of mobilizers recorded more meetings and participants than the other districts. Manhiça district registered fewer activities, which may be related to the low population density in Calanga and Ilha Josina. Activities were also influenced by access. Calanga and Ilha Josina were often affected by severe weather and flooding making the area difficult to access during rainy season [4], therefore during lengthy periods activities relied on a few local facilitators who were able to reach a limited number of target groups. In addition to the challenges of access for community engagement facilitators, the barriers of challenging terrain and poor road infrastructure may also impact access to maternal health services for women [4].

While CE and community partnerships are conceptualized as essential precursors to sustainability of global health projects [22], less is known about the sustainability of CE efforts. Our research shows that it is possible to integrate expanded maternal health messaging into existing health system personnel and routine care, which would support sustainability. However, this may also present some limitations on the delivery of some topics. Because nurses and CHW mostly focused on clinical-oriented topics, the danger signs of pregnancy, especially pre-eclampsia and eclampsia, identification of skilled birth attendants and permission for women to seek medical care were most frequently discussed messages during CE meetings. The reliance of activists' on the capacity of community leaders to mobilize people for meetings and the intermittent presence of the CISM team in the field may have led to lower coverage of the other topics with socio-cultural elements.

CE activities also highlighted the importance and challenges of reaching husbands, partners, mothers-in-law and community leaders that are influential in pregnant women's care-seeking decision making. Pregnant women in previous research in these communities shared that they are expected to obtain permission for seeking maternal health care and delays in seeking care occurred when complications arose and husbands were unaware of the warning signs of pregnancies [16]. A study from Ethiopia found that men's knowledge of obstetric danger signs was poor but there was significantly higher levels of birth preparedness among men who knew at least one danger sign [23]. This supports results of two systematic reviews that found male involvement was associated improved utilization of maternal health services, such as improved antenatal attendance, higher likelihood of facility birth, skilled birth attendant and postpartum care, as well as better birth preparedness as maternal nutrition [24,25]. A locally contextualized CE strategy in Pakistan that engaged both men and women supported increased pre-eclampsia knowledge that pregnant women in these communities had around

seizures and high blood pressure [26]. This research helped frame the CE strategy to engage men but results suggest that there may be contextual challenges for men to participate. Considering the quantitative and qualitative findings together, both datasets highlight the challenge of engaging husbands and partners. The descriptive statistics show the lower numbers of men who participated in engagement activities, while qualitative feedback found that some men questioned why study activities focused on pregnant women and excluded male health concerns. Lower participation of men may be influenced by competing demands. Men in these communities often work long hours outside their homes, and may be required to relocate to Maputo or South Africa for employment, therefore are usually overlooked in activities related to health information, communication and engagement. Future community engagement initiatives should not only invite men to maternal health discussions, but also identify barriers to their participation and reflect on best practices to improve their participation.

### Limitations and strengths

As with any research, there are limitations to our findings. The data were collected in three districts of Maputo and Gaza Provinces; although these results show a good representation of the region, results are not generalizable to other settings. Another limitation was relatively low participation of some groups (decision-makers) and some of the participants were reluctant to share their opinions during the meetings. Geographical challenges such as poor terrain and remoteness of some communities also influenced the reach and pace of CE activities. Additionally, qualitative data was sourced from CE facilitator feedback thus there may be some bias to reporting positive results. While reoccurring participant engagement was encouraged to reinforce key messages and repeated attendance could suggest appreciation of continued community engagement efforts, the anonymization process of CE logs in which names were not collected limits assessment of repeated attendance. Additionally, we designed the community engagement strategy and its assessment to investigate the contribution of the combined efforts of multiple CE approaches within a complex intervention. This is a limitation because it did not allow differentiations explained across the different types of meetings and methods used to engage community members. Lastly, because the purpose of this manuscript was to describe the development and implementation of the community engagement strategy, we focused on descriptive statistics to describe the process. Follow-up is needed to explore the impact of CE on clinical outcomes.

Strengths of these findings include the use of both qualitative and quantitative data to understand the large scale CE effort. The research team had support of leaders at all levels, including involvement of the Mozambique Ministry of Health.

### Conclusions

Our CE strategy in southern Mozambique integrated maternal health promoting activities in routine care of community health workers and nurses and the involvement of community leaders maximized coverage of messages among the target groups. The use of a recognized local structure (community leaders, nurses, CHW and local activists) allowed greater acceptance of CE activities and understanding district differences, engaging household decision-makers and community leaders to build local support for maternal health and flexibility to tailor messages to local needs were important in developing sustainable forms of community engagement. While the current paper demonstrates the effective implementation of a locally developed CE strategy based on maternal health issues identified in the communities, further research is needed to understand potential impacts of CE on maternal health outcomes and strategies to effectively engage men in maternal health promotion.

## Acknowledgments

We would like to thank the men and women in the rural communities of Mozambique where CLIP was conducted and the community engagement facilitators. Additionally, we would like to thank those who collected the data and made this research possible. The CLIP Mozambique Working Group includes: Ana Ilda Biz, Rogério Chiaú, Silvestre Cutana, Paulo Filimone, Marta Macamo, Sónia Maculuve, Ernesto Mandlate, Analisa Matavele, Sibone Mocumbi, Dulce Mulungo, Ariel Nhancolo, Cláudio Nkumbula, Vivalde Nobela, Rosa Pires, Faustino Vilanculo, Rahat N Qureshi, Sana Sheikh, Zahra Hoodbhoy, Imran Ahmed, Amjad Hussain, Javed Memon, Farrukh Raza, Mrutunjaya B Bellad, Shivaprasad S Goudar, Ashalata A Mallapur, Shashidhar G Bannale, Umesh S Charantimath, Keval S Chougala, Richard J Derman, Vaibhav B Dhamanekar, Narayan V Hoonungar, Anjali M Joshi, Namdev A Kamble, Chandrasekhar Karadiguddi, Geetanjali M Katageri, Avinash J Kavi, Gudadayya S Kengapur, Bhalachandra S Kodkany, Uday S Kudachi, Sphoorthi S Mastiholi, Geetanjali I Mungarwadi, Umesh Y Ramadurg, Amit P Revankar, Olalekan O Adetoro, John O Sotunsa, Kelly Pickerill, Jeffrey Bone, Dustin T Dunsmuir, Tang Lee, Jing Li, Beth A Payne, Domena K Tu, Sharla K Drebit, Chirag Kariya, Mansun Lui, Diane Sawchuck, Ugochi V Ukah, Shafik Dharamsi, Guy A Dumont, Tabassum Firoz, Ana Pilar Betrán, Susheela M Engelbrecht, Veronique Filippi, William A Grobman, Marian Knight, Ana Langer, Simon A Lewin, Gwyneth Lewis, Craig Mitton, Nadine Schuurman, James G Thornton, France Donnay.

## Author Contributions

**Conceptualization:** Laura A. Magee, Peter von Dadelszen, Esperança Sevene.

**Data curation:** Felizarda Amosse, Zefanias Nhamirre, Corssino Tchavana.

**Formal analysis:** Felizarda Amosse, Mai-Lei Woo Kinshella, Helena Boene.

**Funding acquisition:** Laura A. Magee, Peter von Dadelszen.

**Investigation:** Felizarda Amosse, Helena Boene, Sumedha Sharma, Zefanias Nhamirre, Laura A. Magee, Peter von Dadelszen, Esperança Sevene, Marianne Vidler, Khatia Munguambe.

**Methodology:** Laura A. Magee, Peter von Dadelszen, Esperança Sevene, Marianne Vidler, Khatia Munguambe.

**Project administration:** Felizarda Amosse, Helena Boene, Sumedha Sharma, Zefanias Nhamirre, Marianne Vidler.

**Supervision:** Laura A. Magee, Peter von Dadelszen, Esperança Sevene, Marianne Vidler, Khatia Munguambe.

**Writing – original draft:** Felizarda Amosse, Mai-Lei Woo Kinshella, Marianne Vidler.

**Writing – review & editing:** Felizarda Amosse, Mai-Lei Woo Kinshella, Helena Boene, Sumedha Sharma, Zefanias Nhamirre, Corssino Tchavana, Laura A. Magee, Peter von Dadelszen, Esperança Sevene, Marianne Vidler, Khatia Munguambe.

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
