## [Decision Letter · Decision Letter 0]

27 Jun 2022

PGPH-D-22-00007

The Development and Implementation of a Community Engagement Strategy to Improve Maternal Health in Southern Mozambique

Dear Dr. Munguambe,

Thank you for submitting your manuscript to PLOS Global Public Health. After careful consideration, we feel that it has merit but does not fully meet PLOS Global Public Health’s publication criteria as it currently stands. Therefore, we invite you to submit a revised version of the manuscript that addresses the points raised during the review process.

Please carefully consider the constructive feedback from Reviewers 1 and 3.

Please submit your revised manuscript by . If you will need more time than this to complete your revisions, please reply to this message or contact the journal office at globalpubhealth@plos.org. Please include the following items when submitting your revised manuscript:

We look forward to receiving your revised manuscript.

Kind regards,

Hannah Tappis, DrPH, MPH

Academic Editor

Journal Requirements:

1. You indicated that you had ethical approval for your study. In your Methods section, please ensure you have also stated whether you obtained consent from parents or guardians of the minors included in the study or whether the research ethics committee or IRB specifically waived the need for their consent.

2. Please amend your detailed online Financial Disclosure statement. This is published with the article. It must therefore be completed in full sentences and contain the exact wording you wish to be published.

State the initials, alongside each funding source, of each author to receive each grant.

3. Please ensure that the funders and grant numbers match between the Financial Disclosure field and the Funding Information tab in your submission form. Note that the funders must be provided in the same order in both places as well.

4. In the online submission form, you indicated that “The datasets generated during and/or analyzed during the current study are available from the corresponding author on reasonable request.”. All PLOS journals now require all data underlying the findings described in their manuscript to be freely available to other researchers, either 1. In a public repository, 2. Within the manuscript itself, or 3. Uploaded as supplementary information.

5. Please provide separate figure files in .tif or .eps format and ensure that all files are under our size limit of 10MB.

6. The following figures have poor resolution: Figures 1, 2 and 3. Please provide higher resolution versions.

7. All figures and supporting information files will be published under the Creative Commons Attribution License (creativecommons.org/licenses/by/4.0/). Authors retain ownership of the copyright for their article and are responsible for third-party content used in the article. 

Figure 1: please (a) provide a direct link to the base layer of the map used and ensure this is also included in the figure legend; (b) provide a link to the terms of use / license information for the base layer. We cannot publish proprietary or copyrighted maps (e.g. Google Maps, Mapquest) and the terms of use for your map base layer must be compatible with our CC-BY 4.0 license. 

Please upload any written confirmation as an 'Other' file type. It must clarify that the copyright holder understands and agrees to the terms of the CC BY 4.0 license; general permission forms that do not specify permission to publish under the CC BY 4.0 will not be accepted. Note that uploading an email confirmation is acceptable.

Additional Editor Comments (if provided):

Reviewers' comments:

Reviewer's Responses to Questions

**Comments to the Author**

1. Does this manuscript meet PLOS Global Public Health’s publication criteria? Is the manuscript technically sound, and do the data support the conclusions? The manuscript must describe methodologically and ethically rigorous research with conclusions that are appropriately drawn based on the data presented.

Reviewer #1: Yes

Reviewer #2: Yes

Reviewer #3: Yes

2. Has the statistical analysis been performed appropriately and rigorously?

Reviewer #1: N/A

Reviewer #2: No

Reviewer #3: No

3. Have the authors made all data underlying the findings in their manuscript fully available (please refer to the Data Availability Statement at the start of the manuscript PDF file)?

Reviewer #1: Yes

Reviewer #2: Yes

Reviewer #3: Yes

4. Is the manuscript presented in an intelligible fashion and written in standard English?

Reviewer #1: Yes

Reviewer #2: Yes

Reviewer #3: Yes

5. Review Comments to the Author

Reviewer #1: Congratulations to the authors!

This study covers a very important topic in the context of maternal mortality reduction strategy.

I have very few questions for clarification:

1. in the abstract, line 31, you mention as one of the CE message, the information about the trial, and I believe related to that you mention in the Box 1, number 7, the study protocol. I am wondering what kind of aspect were discussed with the community in this regard.

2. Being used a mix methodology one could expect to see more of this combination in exploring the findings to support the conclusions. In my view the manuscript describe very well the process, however I still not clear what was the final impact. There was any change on the identified barriers to community health? there was any attempt to measure them?

Reviewer #3: This paper is written clearly and succinctly overall. It provides process evaluation of community engagement activities related to a larger trial.

Title:

- Includes the word “development” but there is little detail regarding the development of the strategy. Suggesting removing this word or adding more detail about the development of the strategy.

Abstract:

- No specific comments, except use of the word “successful” which I’ll discuss later.

Introduction:

Page 6, line 56 – “suggesting sub-optimal health care seeking behaviors” – it also may suggest lack of availability of services. It would be good to discuss the availability of services briefly.

Methods:

Implementation

- The text indicates that activities took place in three spaces: PHCS, homes of pregnant/puerperal women and in community meetings. Does the data from the “CE logs” include only the community meetings, or all three of these contacts?

- Regarding the validity of the CE logs, I can imagine they might undercount the total participation, if some logs were misplaced or facilitators forgot to record them. However, they might over count, if facilitators felt pressured to show productivity. I’d like the authors to reflect on the validity of the logs and whether anything was done to validate this data source.

- What incentives, if any, were the facilitators given to engage in these extra activities?

The analysis would benefit from the use of appropriate statistical tests and more sophisticated analysis. For example, performing appropriate statistical tests for the information presented in Table 3, to determine if the composition of participants varied by district. It would also be interesting to know if the pattern of meeting attendance varied over time.

Results

- Table 5 – Did the project have goals regarding the number of meetings for each group?

- Figure 2 – this should not be a pie chart since multiple topics were discussed at the same meeting.

- Figure 3 – Change to a two-dimensional style of bar chart because the three-dimensional style doesn’t add any information. I don’t understand how to interpret this information – was the goal to have standard presentation of information across settings? Or was the content expected to be different depending on interaction with participants?

- Page 17, line 238 – note typo in spelling of the word “facility”

- Page 18, line 249 – says the facilitators received “some pushback” from men, and the quotations suggest men were critical of the project; were there other men who gave positive feedback, or was the feedback overwhelmingly negative?

Discussion

- The authors state that the intervention reached about 19000 people in a population of roughly 234000. This works out to just under 5 people per meeting or roughly 2.5 interactions per facilitator per week. At the outset, did they have any goals or projections regarding the level of coverage to achieve? And if so, was this the coverage/workload expected?

- Line 265 – Regarding the sentence that begins, “The focus on dialogue to deliver CE messaging helped to promote participant…” this seems like the authors’ professional opinion rather than something they have demonstrated through their data. I think it’s based on the knowledge that informed the development of the intervention. It’s two ideas – one, that dialogue is better way of communicating (as opposed to more didactic messages, I suppose.) Two, that having the facilitators keep records of their process would improve the intervention. I think the authors should expand this sentence into several sentences refer to the scientific literature that informed this thinking.

- Line 266 – typo – it should say “helped to strengthen”.

- Lines 288-293 –These lines are unclear, and it seems like the authors are introducing new information in the discussion section.

-

- Lines 326-331 – this information belongs more in the methods section.

- Line 338 – should be “activists” instead of “activist”

Conclusions:

Using the word “successful” to describe the intervention seems like an over reach, given that no outcomes are presented, and they don’t describe any specific coverage targets or goals the project was trying to achieve.

6. PLOS authors have the option to publish the peer review history of their article (what does this mean?). If published, this will include your full peer review and any attached files.

**Do you want your identity to be public for this peer review?** For information about this choice, including consent withdrawal, please see our Privacy Policy.

Reviewer #1: **Yes: **Leonardo Chavane

Reviewer #2: No

Reviewer #3: **Yes: **Emma K. Williams

---

## [Decision Letter · Decision Letter 1]

17 Oct 2022

PGPH-D-22-00007R1

The Development and Implementation of a Community Engagement Strategy to Improve Maternal Health in Southern Mozambique

Dear Dr. Munguambe,

Thank you for submitting your manuscript to PLOS Global Public Health. Many points raised by previous reviewers have been addressed, however some concerns remain. After careful consideration, we feel that the current draft has merit but does not fully meet PLOS Global Public Health’s publication criteria as it currently stands. Therefore, we invite you to submit a revised version of the manuscript that addresses additional points raised during the review process.

We look forward to receiving your revised manuscript.

Kind regards,

Hannah Tappis, DrPH, MPH

Academic Editor

Journal Requirements:

1. You indicated that you had ethical approval for your study. In your Methods section, please ensure you have also stated whether you obtained consent from parents or guardians of the minors included in the study or whether the research ethics committee or IRB specifically waived the need for their consent.

Additional Editor Comments (if provided):

Reviewers' comments:

Reviewer's Responses to Questions

**Comments to the Author**

1. If the authors have adequately addressed your comments raised in a previous round of review and you feel that this manuscript is now acceptable for publication, you may indicate that here to bypass the “Comments to the Author” section, enter your conflict of interest statement in the “Confidential to Editor” section, and submit your "Accept" recommendation.

Reviewer #4: (No Response)

2. Does this manuscript meet PLOS Global Public Health’s publication criteria? Is the manuscript technically sound, and do the data support the conclusions? The manuscript must describe methodologically and ethically rigorous research with conclusions that are appropriately drawn based on the data presented.

Reviewer #4: Partly

3. Has the statistical analysis been performed appropriately and rigorously?

Reviewer #4: Yes

4. Have the authors made all data underlying the findings in their manuscript fully available (please refer to the Data Availability Statement at the start of the manuscript PDF file)?

Reviewer #4: No

5. Is the manuscript presented in an intelligible fashion and written in standard English?

Reviewer #4: Yes

6. Review Comments to the Author

Reviewer #4: Thank you for this important paper. Well done on the work. It was very interested to read. I have 9 comments that I suggest could strengthen the paper.

Comment 1: The paper lacks a structure/a frame that could help to achieve the expressed aims of the papers: to describe the development of the CE strategy and implementation of this CE intervention.

I suggest setting a structure to describe the development of the CE Strategy that can be further to used to set out the implementation process also.

Table 2 sets out the findings from the feasibility study and how each informed 'community actions'. There are 5 actions in total. Later in the paper does not refer to these again, and rather speaks to activities (and it seems that activities is referring to 3 different types of meetings undertaken to impart key messages) but there is not a clear flow from the content of the CE strategy – the community actions you cite in Table 2 and the ‘community activities’ you proceed to speak to. It will be important to elicit that more clearly.

I would suggest reworking Table 2 to include the fuller activity descriptions. This revision to Table 2 will help to capture the conceptual journey taken from identifying barriers, to needs to defining the activities in the CE Strategy in consultation to determining the activities required to address each. Thereafter, the paper can describe the implementation of activities to meet each of these 5 needs (I note 5 – the transport scheme is in another paper and the table can capture that accordingly).

For example:

LOCAL BARRIER:poor awareness of maternal health issues; pre-eclampsia

IDENTIFIED NEED (IN CONSULTATION WITH COMMUNITIES, CHW, MOH ETC)

Develop and promote educational messages and visual materials about pregnancy warning signs and care seeking behaviours

COMMUNITY ENGAGEMENT ACTIVITIES

1. 48 CHW engage in health promotion using purpose made materials in house visits for women and husbands with 48 CHW

2. 11 nurses engage in PHC health talks, using purpose made materials with women and community members led b

3. 5 activists/mobiliers facilitated community meetings supported by community leaders

Comment 2: Provide more depth in the description of implementation

I would suggest that you extend your analysis to breakdown the numbers you share (In table 4 and 5) further in order to provide a clearer depiction of the implementation of the activities /meetings

• All 'meetings' are aggregated and there is no differentiation explained across the different types of meetings.

Details are provided that there are 3 different spaces/types of meetings; 16 nurses at PHC, 48 CHW in home visits and 5 activist/mobiliser led community meetings. However, these are lumped together as ‘meeting’ in the tables, analysis and discussion.  Can you include a breakdown in the numbers per the type of meeting? 19000 'meetings' is too aggregated and denies an individual analysis of the different activities/meetings.

Currently the analysis does not break down by meeting types and the numbers of participants reached through the different types of meetings, and therefore misses the opportunity to drill down to describe the implementation of these as individual activities. For example - these appears to be zero (or close to zero) reporting on the number of household visits, if they were repeated or one house visit only per household, the participants in the household visits, and feedback during individual house visits with CHW which is unfortunate as this could identify if there are differences in the reflections of individuals when in their private sphere rather than a public sphere.

Stakeholder reflections from Line 247 – appears to represent the reflections form the group formats only.

The high level of aggregation of the presentation of the data means we cannot tell did husbands attend only in households or also in community meetings and PHC talks, which activity/meetings did leaders attend etc.

Comment 3: How the participants are counted.

19169 - is there double counting of those participants that attended recurring meetings, or recurring house visits? This is not clear. If the data that is set out in Table 3 is for individual participants, that should be specified.

• Did participants return to recurring group meetings and was that captured and analysed? Repeated attendance could be an indicator of appreciation for the CE sessions.

Comment 4: There is 0.1% leader involvement overall. Was there a leader at all community meetings? Or at x% of public health talks etc. Were they involved in other activities?

Comment 5: For husbands – 0.9% reported overall. There is an explanation that husbands are working away so participation is low. But there is no description of the strategy devised during the Community Engagement Strategy development, and course correction during implementation to try to improve health promotion to husbands, as the key decision maker.

Comment 6: Use of community terms, definitions and outdated references

The references used to define CE and the critical need for more research are from 2002 and 1991. The literature has developed substantially over the past 10 years in particular. I suggest you use a more up-to-date definition and call for action - delete reference 10 which undermines the strength of this paragraph (this is a book review 1 page from 1991 - better to go straight to WHO /UNICEF documents).

1.     Community engagement; a health promotion guide for universal health coverage in the hands of people. Geneva. World Health Organization 2020.

2.     WHO Recommendations on health promotion interventions for maternal and newborn health

• Also, the literature on community engagement has used different terminology over time and as a result there are multiple terms in use and the terms are used interchangeable; community engagement, community participation, community mobilisation etc. Three terms are used in this paper, interchangeably. In order to avoid adding to the confusion in the literature, I would suggest that to stick with the use of one term – community engagement and define that.

• CE has many definitions, and the scope can entail health promotion, health education, citizen involvement in quality and coverage improvement through M&E activities, collaboration in health facility management, community health, monitoring, dialogues with health authorities on needs etc. The purpose of the activities set out in this paper is engaging community members firstly in health promotion and awareness raising on danger signs, how to seek referral. The further activity on developing a group saving scheme is not described and I understand this is for another paper. Therefore, I would suggest you draw on an up-to-date definition that is tailor-made for community engagement for the health promotion activities that this paper is focused on: such as WHO Recommendations on health promotion interventions for maternal and newborn health (2015).

Comment 7: There is little analysis provided on the views and feedback of the implementers themselves during the implementation or description of how this resulted in course correction, changes, how any challenges faced were reviewed and overcome. Line 301: Highlights the value of facilitator reflection– but there is no description or examples provided of the content the facilitators reflections - the nurses and CHW in particular? It would be good to share both positive and negative reflections that arose during implementation. What were the challenges and disaggregate these such as for the challenges the nurses faced at the PHC, the CHW on house visits etc., was there a process followed for implementers to reflect, tweak the activity /change course etc during implementation?

Comment 8: The paper notes that facilitators (chiefly the nurses and CHW) reported the activities were well received by the community. This is a key finding. This could be bias due to the facilitator interests to appear to a good job etc – and it would be better if there was further content analysis undertaken such as on evaluation interviews with participants, community leaders, CHW, nurses to substantiate the finding.  As it is - it would be good to qualify that statement to be clear that that finding is based on the personal views of facilitators that the activities were well received.

Comment 9: Overall need to check for typos – for example.

Abstract: 2107 - 2017. Again in Line 28

Line 53 – contributes

Line 54 – services

Line 61 – suggest change management of mortality to reduction of maternal and infant mortality.

7. PLOS authors have the option to publish the peer review history of their article (what does this mean?). If published, this will include your full peer review and any attached files.

**Do you want your identity to be public for this peer review?** For information about this choice, including consent withdrawal, please see our Privacy Policy.

Reviewer #4: No

---

## [Editor Report · Decision Letter 2]

4 Jan 2023

The Development and Implementation of a Community Engagement Strategy to Improve Maternal Health in Southern Mozambique

PGPH-D-22-00007R2

Dear Dr. Munguambe,

We are pleased to inform you that your manuscript 'The Development and Implementation of a Community Engagement Strategy to Improve Maternal Health in Southern Mozambique' has been provisionally accepted for publication in PLOS Global Public Health.

Best regards,

Hannah Tappis, DrPH, MPH

Academic Editor